Integrating cyber-physical systems with embedding technology for controlling autonomous vehicle driving

Alohali Manal Abdullah 1
Alqahtani Hamed 2
Darem Abdulbasit 3
http://orcid.org/0000-0002-9713-4296 Abdullah Monir 4
http://orcid.org/0000-0002-3318-9394 Nam Yunyoung 5 ynam@sch.ac.kr
Abouhawwash Mohamed 6 7
1 Department of Information Systems, Princess Nourah bint Abdulrahman University , Riyadh , Saudi Arabia
2 Department of Information Systems , Abha , Saudi Arabia
3 Department of Computer Science, Northern Border University , Arar , Saudi Arabia
4 Department of Computer Science and Artificial Intelligence , Bisha , Saudi Arabia
5 Department of ICT Convergence, Soonchunhyang University , Asan , Republic of South Korea
6 Department of Computational Mathematics, Michigan State University , East Lansing , United States
7 Department of Mathematics, Mansoura University , Mansoura , Egypt
Stević Željko
Electronic publication date: 2025 Jun 10
Publication date: 2025
Volume: 11
Electronic Location ID: e2823
Received 2024 Oct 22; Accepted 2025 Mar 21
Copyright: © 2025 Alohali et al.
Copyright year: 2025
Copyright holder: Alohali et al.
License: This is an open access article distributed under the terms of the Creative Commons Attribution License, which permits unrestricted use, distribution, reproduction and adaptation in any medium and for any purpose provided that it is properly attributed. For attribution, the original author(s), title, publication source (PeerJ Computer Science) and either DOI or URL of the article must be cited.
License URL: https://creativecommons.org/licenses/by/4.0/

Keywords: Autonomous vehicles, Cyber-physical systems, Reinforcement learning, Deep Q networks, Embedded technology

Funding: Korea Institute for Advancement of Technology (KIAT) grant funded by the Korea Government (MOTIE) P0012724 National Research Foundation of Korea (NRF) grant funded by the Korea government (MSIT) RS-2023-00218176 Soonchunhyang University Research Fund The Deanship of Research and Graduate Studies at King Khalid University funded this work through Large Research Project RGP2/267/45 Princess Nourah bint Abdulrahman University Researchers Supporting Project PNURSP2025R330 Princess Nourah bint Abdulrahman University, Riyadh, Saudi Arabia The Deanship of Scientific Research at Northern Border University, Arar, KSA NBU-FFR-2025-2903-04 The Deanship of Graduate Studies and Scientific Research at University of Bisha supported this work through the Fast-Track Research Support Program This research was supported by the Korea Institute for Advancement of Technology (KIAT) grant funded by the Korea Government (MOTIE) (P0012724, HRD Program for Industrial Innovation), the National Research Foundation of Korea (NRF) grant funded by the Korea government (MSIT) (No. RS-2023-00218176), and the Soonchunhyang University Research Fund. The Deanship of Research and Graduate Studies at King Khalid University funded this work through Large Research Project under grant number RGP2/267/45. Princess Nourah bint Abdulrahman University Researchers Supporting Project number (PNURSP2025R330), Princess Nourah bint Abdulrahman University, Riyadh, Saudi Arabia. The Deanship of Scientific Research at Northern Border University, Arar, KSA funded this research work through the project number “NBU-FFR-2025-2903-04”. The Deanship of Graduate Studies and Scientific Research at University of Bisha supported this work through the Fast-Track Research Support Program. The funders had no role in study design, data collection and analysis, decision to publish, or preparation of the manuscript.

==============================
Cyber-physical systems (CPSs) in autonomous vehicles must handle highly dynamic and uncertain settings, where unanticipated impediments, shifting traffic conditions, and environmental changes all provide substantial decision-making issues. Deep reinforcement learning (DRL) has emerged as a strong tool for dealing with such uncertainty, yet current DRL models struggle to ensure safety and optimal behaviour in indeterminate settings due to the difficulties of understanding dynamic reward systems. To address these constraints, this study incorporates double deep Q networks (DDQN) to improve the agent’s adaptability under uncertain driving conditions. A structured reward system is established to accommodate real-time fluctuations, resulting in safer and more efficient decision-making. The study acknowledges the technological limitations of automobile CPSs and investigates hardware acceleration as a potential remedy in addition to algorithmic enhancements. Because of their post-manufacturing adaptability, parallel processing capabilities, and reconfigurability, field programmable gate arrays (FPGAs) are used to execute reinforcement learning in real-time. Using essential parameters, including collision rate, behaviour similarity, travel distance, speed control, total rewards, and timesteps, the suggested method is thoroughly tested in the TORCS Racing Simulator. The findings show that combining FPGA-based hardware acceleration with DDQN successfully improves computational efficiency and decision-making reliability, tackling significant issues brought on by uncertainty in autonomous driving CPSs. In addition to advancing reinforcement learning applications in CPSs, this work opens up possibilities for future investigations into real-world generalisation, adaptive reward mechanisms, and scalable hardware implementations to further reduce uncertainty in autonomous systems.

Introduction

The subsequent significant development in the relentless advancement of digital technologies is cyber-physical systems (CPS) (Eskandarian, Wu & Sun, 2019). The world is adopting CPS quickly, and sophisticated cyber-science and physical processes are integrating rapidly. Numerous processes in physical systems occur in tandem with their inherent uncertainty. Comprehensive computing skills are needed to evaluate, observe, and manage the fluctuating behaviour of physical systems leveraging cyber systems (Damaj et al., 2021). These skills are also required to integrate connections and processing, generate real-time choices, and implement suitable assessment mechanisms. These days, with significantly improved computer power and advanced artificial intelligence, it is feasible to get an in-depth understanding of the physical systems that need to be tracked and to create more reliable, precise, adaptable, and economical solutions. Cyber-physical systems coordinate the digital and tangible realms (Xiao, Guo & He, 2021).

The collaboration that results from this amalgamation will profoundly change the future of human-engineered system interaction. The administration of resources, autonomous vehicles, company administration, energy planning, learning, business, innovative manufacturing, and sustainability monitoring are a few areas where the emergence of CPSs is supplying, restructuring, and generating multiple novel methodologies (Poudel & Munir, 2018). CPSs have great potential to revolutionise the transportation industry in the future because of the smooth integration of several intricate relationships between ubiquitous mechanical operations and intelligent data processing (Deng et al., 2021).

The current research will develop improved characteristics and adaptability to encompass sophisticated algorithms-based virtual conceptions and physical processes. However, since the interconnected systems have become complicated, research is still needed to ensure the systems’ dependability, controlled activities, productivity, and maintenance—despite enormous achievements in creating CPSs (Vasudev et al., 2020). CPSs guarantee that the world will be better with reliable and effective systems. Nevertheless, one of the most significant research issues is the security of the emerging cyberspace. Furthermore, there are additional risks associated with the widespread adoption of CPS in the transportation sector because solely motorized parts have a lower failure rate than a system of actuators as well as sensing devices, which is much more vulnerable to malfunctioning hardware, software inaccuracies, and hacking attempts (Chattopadhyay, Lam & Tavva, 2020).

The cyber-physical system for the automotive industry is multifaceted and comprises several physical elements such as motor vehicles, road structures, automobile drivers, industrial equipment, and sensors for analysis (Gazis, Ioannou & Katsiri, 2019). Specifically, because vehicles travel at such rapid paces, the physical layout of the automotive mass transit system is constantly changing. Furthermore, because motorists can join or leave the infrastructure at any time, their actions have immediate consequences on the configuration of the network. Depending on the area, the automotive occupancy within the vehicular CPS also changes (Novakazi et al., 2021). Automotive networks also stand out from other forms of wireless networking due to their practically limitless power, retention, and ability to compute. Furthermore, most communication devices used in everyday life were not created with swiftly evolving automobiles in mind.

Low latency is essential for security-relevant applications in the automotive CPS that need urgent notifications sent promptly (Duan et al., 2021). However, bandwidth constraints occur, mainly when electronic media and information systems transmissions are considered. The construction of intelligent systems that can manage many intelligent sensors integrated into automotive CPS is made possible by ongoing advancements in artificial intelligence (AI) as well as machine learning (ML) technologies (Zheng et al., 2020). These systems enable failure estimation, increased independence, and autonomous algorithms for more trustworthy and secure deployments. Moreover, embedded AI techniques can maximise networked service providers’ transmission speed and power usage, improving system effectiveness and reliability (Noh, An & Han, 2015). Therefore, a robust development platform for automotive CPS needs to facilitate the deployment of sophisticated AI algorithms along with embedded technology (DiPalma et al., 2021).

Trustworthiness, stability, and durability are essential for every system and are crucial to safety. Unpredictable behaviour in driverless cars can potentially cause fatal crashes and severely harm the surrounding infrastructure. However, as technology advances, a wider range of gauges, layouts, and CPS characteristics are becoming available (Tang et al., 2021). Determining a system’s compatibility and consistency is challenging without conducting multiple, conceivably costly inbound evaluations. Furthermore, one major obstacle to implementing cyber-physical systems in the automotive industry is the requirement to effectively model complex structures (Chattopadhyay & Lam, 2017). Therefore, it is necessary to provide a single set of cyber-physical guidelines covering various scenarios and operations (Ling et al., 2020).

AI algorithms may make poor conclusions when not adequately trained, which adds to the ambiguity. This challenge is more severe in cases where autonomous vehicles depend on outside facilities for navigation because roadway markings and symbols frequently change from one place to another and tend to become obsolete with time (Shin, Park & Park, 2020). Therefore, it must become more deterministic and explicable to facilitate AI’s employment in applications where safety is paramount. The main problem with the real-world autonomous driving scenario is that it comprises unpredictable environmental elements (Khatun, Glass & Jung, 2021). The actual environment in which autonomous driving operates is complex, making it possible to disregard some indeterminate aspects of the environment. This implies that the intelligent driving system may be unreliable because an identical action taken in the same situation may result in an entirely distinct state in the future (Eskandarian, Wu & Sun, 2019). This indicates that the initial strategic choice made by the system that drives autonomously cannot finish the task, and it must be modified depending on the cognitive results of the indeterminate context (Bijlsma et al., 2020).

To overcome the problems mentioned above, a deep reinforcement learning-based method such as the double deep Q network (DDQN) algorithm is employed for creating decision controllers for autonomous driving CPS in an indeterminate context, where anticipated cognitive restrictions direct the learning process. To support this type of ever-more-complex intelligent algorithms, reliable hardware architectures are also necessary to overcome the technological limitations of automotive CPSs. Hence, field programmable gate arrays (FGPA) are chosen as a potentially helpful hardware option to be incorporated in automotive CPS. These devices made from semiconductors can be reconfigured to meet particular usage or operational needs after manufacturing. They are built around customisable logical modules coupled with configurable links. Field programmable gate arrays (FPGAs’) high degree of concurrency makes it possible for them to support faster AI-based algorithm implementations and more accurately reproduce associated physical procedures. These hardware designs can achieve distinct alternatives between enhancing productivity, reliability, and data fidelity and reducing delay and energy usage.

Research contributions

(1) To employ a DDQN algorithm designed for autonomous driving under indeterminate conditions to optimize decision-making processes and improve the safety and dependability of autonomous driving CPS.

(2) To incorporate FPGAs as a hardware solution to overcome the technical barriers of sophisticated AI algorithms, such as DDQN, with enhanced performance measures through high parallelism and adaptability.

(3) To demonstrate the proposed system’s effectiveness, scalability, and feasibility in simulated autonomous driving applications.

Article organization

The remainder of the article is organized as follows. “Related Work” describes the existing works in the literature concerning applications of reinforcement learning for autonomous driving and embedded architectures for such systems. “Proposed Methodology” presents the proposed methodology, which includes the RL architecture explaining the identification of indeterminate states, the rewarding mechanism adopted, the DDQN algorithm as a decision controller, and the FPGA-driven embedded framework. “Results and Discussion” presents the simulation results and evaluates the efficiency of the proposed approach. “Conclusion” concludes the present research.

Related works

This section describes the current studies conducted by various researchers in automotive cyber-physical systems and their applications using AI algorithms and embedding technology. Guo et al. (2020) stressed that the expanding interconnectedness of automotive CPS brings about substantial confidentiality and safety issues. The study suggests that it is essential to ensure that the CPS structure is precise, dependable, safe from cyberattacks, and secure as they become more widely used. Notwithstanding these dangers, automotive CPS is the next big thing in technology and has the power to completely change the way we travel, reside, and perform tasks in the future. Researchers in Fayyad et al. (2020) state that improving transportation infrastructure in smart communities requires integrating the Internet of Things (IoT) with automotive CPS. The authors contend that by creating an innovative and cooperative transportation network, autonomous automobiles can significantly improve the effectiveness of roadway management systems, ultimately leading to enhanced highway security.

Similarly, investigators in Sabaliauskaite et al. (2018) attest that real-time intelligent vehicles protect susceptible users of the road by utilizing increased eyesight, radar detectors, satellite systems, catastrophe alert devices, racing signals, and protection markers. Additionally, they highlight how these systems can be used to forecast traffic statistics, chauffeur, and customer actions, which helps to improve traffic control and scheduling algorithms. The authors also discuss how intelligent systems are essential for operating electric vehicles with different levels of autonomy, such as entirely or partially autonomous vehicular systems and supported navigation. In addition, Rasheed et al. (2020) provide an industrial proposition for autonomous cars with interaction facilities between the infrastructure and automobiles. It has been concluded that using a collaborative transportation system is more economical and efficient than depending on complete autonomy.

One emerging study area is the security management of automotive CPS systems, where reinforcement learning (RL) is a hot topic. A constrained maximum efficiency parameter is used throughout numerous security-based reinforcement learning techniques. The agent looks for already established acceptable control behaviors throughout the policy domain. Efficiency parameter was adjusted by Shawky et al. (2022) to consider the security concern, giving the agent some initial information to steer strategy development away from potentially hazardous circumstances. Understanding the difference between investigating current control alternatives and optimizing within established security standards is essential.

In security-based RL methods, reward stability and security protocol verification are two essential evaluation metrics. Reward stability ensures the learning system’s objectives align with human intents (Sharath & Mehran, 2021). This is so that RL agents can benefit from the incorrect objectives. The understanding of the reasons behind why learned policies make specific decisions is the aim of security protocol verification. This is essential for the safe deployment of RL agents. Reward stability allows the task’s objective to be represented as reward-dependent or independent. One common technique proposed for creating powerful reward functions is reward structuring.

Campisi et al. (2021) employed a statistical framework for reinforcement learning in a volatile setting to verify a validation approach based on stochastic chronological reasoning constraints; they then modelled the environment as an idealised Probabilistic decision-making system to verify the security protocol. On the other hand, few research (Aldakkhelallah & Simic, 2021) consider the worst-case scenario under ambiguity, producing a control plan following target optimization to maximise reward.

The research in Ali et al. (2021) presented a reinforcement learning system that minimises the intended hazard and determines the best driving strategy by integrating a hazard-evaluation function. However, the authors did not consider unpredictable environmental elements, including operating fashion, and only evaluated driving risk regarding location ambiguity and distance-based precautions. Therefore, the present research is proposed to characterise operating hazards associated with unpredictable environmental elements. Zhu et al. (2021) added several improvements to the reward function adaptation technique. Using an enhanced reward system, they balanced accomplishment rewards from collaborative and aggressive approaches, incident extent, and passenger convenience. They did not, however, consider the impact of variations in velocity and operating technique; instead, they focused on the scenario involving several crashes of vehicles.

This allows vehicles to communicate and transmit data such as whereabouts, velocity, and direction, which can then be used for avoiding collisions, managing traffic, and improving driving safety (Sun et al., 2019, 2018a, 2018b). This functionality is critically important in enhancing the comfort and effectiveness of public transport since it alerts the commuters regarding the approximate time of arrival, thereby minimizing uncertainty, and helping them plan their journeys better (Rong et al., 2022; Lv, Ju & Wang, 2025). In any dynamic system, oscillations are unwanted because they contribute to inefficiency, instability, or unsafe conditions.

Owing to oscillation, loss of precision or deviation from a pre-set path is possible within robots (Li et al., 2023b; Ju et al., 2023). Spatio-temporal prediction being a complex task is greatly reliant on the ability to capture both local and global dependencies within a long sequence of data, an action in which these mechanisms ontogenetically excel (An et al., 2024; Wang et al., 2024a). While these vehicles perform tasks such as object detection, planning the route, and making decisions, the amount of computations needed to accomplish such tasks can be too much for the on-board systems to handle (Yao et al., 2023; Peng et al., 2024). Use multiple sensor types to obtain and monitor a wide range of measurements of the vehicle and its environment (Song et al., 2024; Wang et al., 2024c). In vehicle identification, segmentation assists in the extraction of image regions that contain vehicles from non-vehicle regions (Li et al., 2023a; Wang et al., 2024b). Path-tracking is considered one of the most significant functions of autonomous cars and it is accomplished with the help of a controller that is capable of responding to many dynamic factors such as a changing road or speed (Yang et al., 2023; Mohammadzadeh et al., 2024). In this case a hybrid model is a model of lateral motion dynamics of the vehicle built using traditional methods, neural networks, and other modern methods of machine learning (Zhou et al., 2023, 2024).

Acquiring vast datasets for model training is a challenging and costly task due to vehicle dynamics in real life applications (Chen et al., 2024a; Qi et al., 2024). Enables the rapid dissemination of vast amounts of information between vehicles, traffic control centers, and infrastructure which allows making smarter decisions in real time (Cai et al., 2023; Ding et al., 2024a). This control is done through an event where significant changes in the environment or the system occur as opposed to constantly tracking every erminal detail (Ding et al., 2024b; Chen et al., 2024b). The goal is to reach consensus: an agreement among all agents in the system regarding a particular value or decision like position, speed or a required task (Yue et al., 2024; Ji et al., 2024). These systems comprise a network of vehicles or other smart devices capable of sharing information for collaborative services such as traffic control, accident prevention, or environmental surveillance (Liang et al., 2024; Xiao et al., 2024). This method also employs RANSAC (random sample consensus), a robust statistical technique that increases positioning accuracy and reliability with noisy or outlier data (Cheng et al., 2024; Xu et al., 2025). In networked systems, some data packets are sometimes lost due to network problems. “Bounded” means the packet dropout rates are bounded above and below by known values (Zeng et al., 2024;, Fu et al., 2024).

In urban areas and congestion, occlusions could greatly hamper an autonomous vehicle’s decision making ability which could result in safety issues and navigation difficulties (Hu et al., 2023; Xiao et al., 2023).

Large datasets often draw insights from a variety of driving environments like urban, suburban, and rural settings, as well as diverse road conditions (Guo et al., 2023; Li et al., 2019). Mobile traffic prediction can be done through an spatial knowledge graph (SKG) which can represent nodes such as road segments, intersections, traffic lights, vehicles, and other relevant entities (Gong et al., 2023; Luo et al., 2022).

Motivation of current research

Much research on automotive CPS indicates several gaps that must be addressed. The fundamental integration of physical and cyber design concepts into a unified design approach is vital. This integration must be supported by dependable legislative, acceptable, and regulatory frameworks that facilitate the accreditation of AI for enhanced independence. To promote its deployment in critical areas of safety like driverless aircraft and autonomous automobiles, machine learning algorithms also need to become more dependable. Similarly, developments in multiagent cooperation, sustainable processing, and swarm cognitive computing will make integrating several cyber-physical agents into a coherent automotive CPS with dynamic and resilient characteristics easier.

This work presents an easy-to-use yet reliable platform for exploring the various components of automotive CPS appropriate for novice and expert users. In an FPGA-driven embedded framework, the architecture facilitates the incorporation of several detectors, transmission lines, and cognitive interfaces. This makes the entire system compatible with a range of hardware platforms and enables the implementation of intelligent deep reinforcement algorithms at embedded and interconnected levels.

Proposed methodology

The proposed methodology uses a deep reinforcement learning-based technique like the DDQN algorithm to develop precision controllers for autonomous driving CPS in an unknown environment. In this scenario, the learning process is guided by expected cognitive constraints. To overcome the computational limits of automobile CPSs, dependable hardware architectures are also required to handle this kind of complicated, intelligent algorithm. Therefore, it has been decided to include a FGPA-driven embedded framework as a possibly beneficial hardware alternative in automobile CPS. The structure of the proposed system is presented in Fig. 1.

Figure 1 Proposed system.

Problem formulation

Markov decision processes (MDPs) can be optimised for rewards over the long run using the reinforcement learning (RL) technique, which is widely used in artificial intelligence. During the training of an RL model, an agent lacking any experience should continuously explore its surroundings to gather data, and it must repeatedly attempt to determine the best course of action. During the agent’s learning procedure, the surroundings modify their current state according to each agent’s provisional choices. Then the agent is awarded a reward value based on the altered condition. The agent optimizes its approach throughout persistent discovery, keeps evolving, and ultimately determines the best choice strategy based on ongoing observations of surroundings and responses received.

A Markov decision process is considered to be a combination of the agent’s varying states ( SP), actions to be performed ( AP), function denoting the transition between states ( TFP) and rewards given for successful actions ( RFP) that are accomplished. It can be mathematically represented as shown in Eq. (1),

(1) MDP=(SP,AP,TFP,RFP).

The main objective is to apply the most optimal technique that can enhance the likelihood of obtaining efficient outcomes as denoted in Eq. (2),

(2) γ=argmaxγ(EV[∑k=0K−1rf(SPk,APk,SPk+1)]).

The above equation is applicable for k number of iterations executed for a particular policy to obtain the maximum value of returns as denoted in EV. The value of the optimised objective ( OPTγ) for autonomous driving using automotive CPS with states and corresponding actions updated can be represented as shown in Eq. (3),

(3) OPTγ(SPk,APk)=EV[∑k=0K−1rf(SPk,APk,SPk+1)SP0=SP,AP0=AP].

RL-based architecture

The architecture is categorised into the physical and the simulated environment for autonomous driving scenarios using automotive CPS. The indeterminate aspects of the autonomous driving scenario are initially obtained using the simulated environment. The identified factors are modelled in the consequent step by considering the indeterminate scenarios. The model comprises a function to generate rewards for the agent’s successful actions. An observer function is also defined to keep track of the outcomes produced for every action the agent performs. This observer function is essential to satisfy the security requirements of the RL-based model. Finally, the inputs from the simulated environment are passed onto the physical setting for executing the tasks.

Identifying indeterminate states

In autonomous driving, unpredictable and dynamically changing settings that influence decision-making are indeterminate contexts. These indeterminate contexts can be mathematically modelled as a partially observable Markov decision process (POMDP) as shown in Eq. (4)

(4) POMDP=(Ps,Pa,Pt,Pr,Po,H,μ).

In the above definitions, Ps refers to the collection of possible states in the environment, Pa refers to the set of all possible actions that may be executed by an agent, Pt refers to the probability of transitioning from state s to s′ to action a, Pr refers to the reward function defined on states and actions, Po refers to the set of all possible observations, H refers to the probability of having observation o for every new state s′, and μ is the discount value for any reward that will be delivered in the future. Given that nature is not defined, Pt is non-deterministic which means external factors influence state changes.

In the above equation, Ps denotes the set of possible states in the environment, Pa denotes the set of potential actions that can be taken by an agent, Pt denotes the probability of transition from state s to s′ for action a, Pr denotes the reward function based on the states and the corresponding actions, Po denotes the set of all possible observations, H denotes the probability of making observation o for any new state s′, and μ denotes the discount factor for any reward to be given in the future. Since the environment is indeterminate, Pt is non-deterministic, which means outside disturbances affect state transitions.

Indeterminate effects on autonomous driving systems come in three forms: detector, exterior, and interior instability. The terms “interior instability,” “exterior instability,” and “detector instability” refer to variations in the data that the detector acquires, the system’s overall surrounding conditions, and the system’s inherent operations. The exterior indeterminacy needs more focus since it occurs due to the forces surrounding the system, and the interior and detector instabilities are influenced mainly by their designs. Exterior ambiguity consists of elements such as the position of vehicles and the climatic conditions. The climate category is affected by the degree of humidity, moisture, heat, rain, and other variables. The unpredictable driving direction of the nearby vehicles is the other factor in the exterior ambiguity. The indeterminate environment, in which interactions with other vehicles play a role, frequently influences the choices that lead a vehicle to change lanes. Figure 2 depicts the scenarios of exterior ambiguity conditions.

Figure 2 Exterior ambiguity scenarios.

(A) Exterior ambiguity due to the position of vehicles: This segment explains that the dense clustering and layout of vehicles on a road can create visual confusion for detection systems. (B) Exterior ambiguity due to climatic conditions: This segment explains how an effect of weather like rain, fog, or glare creates challenges in seeing and intricacies in detecting images using sensors.

The likelihood of determining the driving position of the nearby vehicles is computed using the Eq. (5) based on the values of weights involved to train the model ( wk) and coordinated of current positions ( Sak),

(5) Posv=∑k=1Nwk×Sak.

Rewarding mechanism

In reinforcement learning, the reward function is typically viewed as a “black box” with no means of deciphering its configuration. With this method, the agent can only choose the outcome and is unaware of the state-change connections or contextual transmission modifications. For the agent to gain knowledge from interactions with its surroundings, it is essential to appropriately interpret the rewards from environmental data. Autonomous vehicles should aim to travel as quickly as possible without exceeding the permitted speed threshold to increase effectiveness. As a result, the rewards based on the automobile’s acceleration rate is determined using Eq. (6).

(6) RFspeed=δ1(Speedcurr−Speedsetlimit+ρ1).

In Eq. (6), Speedcurr and Speedsetlimit denotes the current speed of the vehicle and the limit set for the speed of the vehicle respectively, δ1 and ρ1 are used as parametric values to normalize the speed limits. The fact that an autonomous vehicle is often much closer to its neighbor than the safe distance can be a major contributing factor to accidents on the paths. To mitigate this, the reward function of sudden shifts regarding automobile proximity has been developed as shown in Eq. (7),

(7) pdist=(Speedanterior+|Accanterior|Rdelay−Speedposterior)22(|Accanterior|−|Accposterior|)−|Accanterior|Rdelay22.

In the above equation, Speedanterior, Accanterior, Speedposterior and Accposterior denote the speed and acceleration of the vehicles in the anterior and posterior positions respectively. Rdelay denotes the delay in responding to any action. The values of Speed as well as Acc are determined using Eqs. (8) and (9) as,

(8) Speed=pdist(R+ΔR)−pdist(R)ΔR

(9) Acc=Speed(R+ΔR)−Speed(R)ΔR.

A reward is allotted when the agent drives by maintaining proper distance between the other vehicles that are on the same path and the corresponding reward function RFdist is calculated as described in Eq. (10) using the values of pdist and adist which are distance limits and actual distance maintained by the autonomous vehicle with δ2 and ρ2 as parametric values to normalize the distance limits,

(10) RFdist=δ2(pdist−adist+ρ2).

Further, in order to provide a negative reward when the vehicle changes direction in the wrong path, a function is known as RFillegitimate. If these directions are accomplished legitimately as per the regulations, a positive reward is given as denoted by RFlegitimate. If, in case, the autonomous vehicle collides with another vehicle, then again, a negative reward is given as represented as RFcrash. When the autonomous multiple unnecessary changes in the path, then a negative reward is allotted as RFirrational. Thus, the rewards given to the automotive CPS during the different states experienced while driving is summarized in Table 1.

Table 1 Summary of reward functions.

Reward no.	Reward name	Reward type	Reward description	
1	RFspeed	Positive	Reward for maintaining the speed of the vehicle	
2	RFdist	Positive	Reward for maintaining the safe distance between vehicles	
3	RFillegitimate	Negative	Reward for wrong path changes	
4	RFlegitimate	Positive	Reward for valid path changes	
5	RFcrash	Negative	Reward for collisions with vehicles	
6	RFirrational	Negative	Reward for increased number of path changes in irrelevant context	

Double deep Q network based decision controller

This section uses a reinforcement learning method based on value functions to choose an action. By fusing deep neural networks with reinforcement learning approaches, double deep Q effectively addresses multidimensional catastrophe-related problems by substituting the results from the neural network for the outcomes from from Q-value. The issue of exaggerating the values of Q in the conventional DQN model is fixed by DDQN, which is developed as the DRL technique.

The outcome value for any state SP+1 is determined by computing the value of action A′ which is considered to be the largest Q value based on weight parameter W as represented in Eq. (11),

(11) A′=argmaxA⁡Q(SP+1,A;W).

The updated representation of the Q Network is as shown in Eq. (12),

(12) Yout=RFP+1+αQ′(SP+1,argmaxA⁡Q(SP+1,A;W);W′).

In the DDQN algorithm, the created estimation target chooses the biggest Q-value in the desired network without considering its association with the present network, which would cause an exaggeration. The gradient descent approach updates the variable values by Eq. (14), while the mean square error method uses Eq. (13) as the loss function for DDQN calculation.

(13) LF=(Yout−Q(SP,A;W))2

(14) W=W−γdL(W)dW.

The target network replicates the present network settings W for every set of iterations to sustain the reliability of the prediction objective for a given duration. As a result, a more robust algorithm for decision controlling is guaranteed by a wider iteration interval of the target network, but with the drawbacks of a lower refresh rate and slower pace of convergence. Setting a suitable refresh interval is, therefore, essential. DDQN algorithms frequently use this parameter substitution technique, in which all of the network parameters are changed in order to enhance the efficiency of the decision controller. The algorithm for DDQN model is presented in Algorithm 1.

Algorithm 1 DDQN decision controller algorithm.

1: Input: Initial network Qβ, Output Network Qβ′, Replay buffer Rb, φ≪1	
2: Output: Optimal state updates for decision making	
3: for each iteration do	
4:  for each input step do	
5:   Perceive state SP and action AP∼δ(AP,SP)	
6:   Implement AP and perceive consecutive state SP+1 and reward RP=R(SP,AP)	
7:   Save (SP,AP,RP,SP+1) in replay buffer Rb	
8:  end for	
9:  for each refresh step do	
10:   Sample Gp=(SP,AP,RP,SP+1)∼Rb	
11:   Determine output Q value as per Eq. (11)	
12:    Yout=RFP+1+αQ′(SP+1,arg⁡maxAQ(SP+1,A;W);W′)	
13:   Execute gradient descent on output Q value as per Eq. (12)	
14:   LF=(Yout−Q(SP,A;W))2	
15:   Perform optimal state updates:	
16:    β′←φ∗β+(1−φ)∗β′	
17:  end for	
18: end for	

FPGA-driven embedded framework

The objective of this work is to create a basic FPGA-driven embedded system for automotive CPS that accurately captures the particular demands of autonomous vehicles and is compatible with different communication as well as navigation technologies. The chosen method makes use of an FPGA development board, which offers a straightforward way to connect different market-ready components with the embedded system. An inertial measurement unit and a rotational encoding device, which stands in for typical detectors and connections seen in autonomous vehicles, are attached to the FPGA. Through a serial asynchronous connection link, the device notifies a monitoring machine of its precise location, direction, and wheel orientation each time the encoding device rotates. After that, an advanced application records and displays incoming information instantly. The structure of this framework is presented in Fig. 3.

Figure 3 FPGA architecture.

Moreover, integrated firmware runs on an embedded microcontroller base with constrained storage. In addition to mimicking the storage and computational limitations of actual embedded devices, this provides users unfamiliar with hardware description languages with a comfortable programming environment. An onboard human machine interface is installed with standard features like a reboot switch as well as LED illumination to guarantee an easy-to-use system. To evaluate methods of communication and security measures, the system also creates a basic data connection between the embedded agent as well as a monitoring machine. This connection can be changed to a more advanced physical or virtual link. The monitoring machine can also be an autonomous manager to implement a complicated automotive CPS with multiple agents, enabling interaction between numerous integrated units.

Results and discussion

This section presents the experimental results of the proposed system by applying it to a simulated environment.

Simulation environment

Hardware setup: Numerous networked elements that communicate with one another make up autonomous vehicles. As such, open-source embedded hardware forms the core of the simulated testbed. The automotive CPS is disintegrated into several parts, such as the computing elements, sensing and actuating devices, and the modelling workstation, which is considered the physical plant. In an interconnected CPS context, transmission is facilitated by a localised network. The local network comprises a megabit-per-second Bus network and a hundred megabits-per-second network over Ethernet to allow authentic automobile layouts. An NVIDIA Jetson Texas model 2 chip is supplied as the computing framework for creating complicated controllers. BeagleBone Black Industrial AM3358 ARM Cortex-A8 integrated in the engine control unit to handle the reduced sophistication transitional sensory and actuation software. An Intel Core i7 personal computer with a 5400 RPM hard drive makes up the simulation workspace.

Software setup: The software design of the simulation environment is the implementation of CPS controlling algorithms for interacting with and managing an autonomous vehicle in a networked model. The TORCS Racing Simulator (Wymann et al., 2000) is the autonomous driving car simulator employed in testbed. This simulator can be used on various operating systems, including Mac, Linux, and Windows. 18.04 LTS is the operating system used to run the simulator. In addition, customised Python-based programming is introduced to facilitate parameter accessibility from external processes. Sensor values, including lidar technology, velocity, accelerator, gear, track location, proximity to the initial position, automobile heading, and location during the race, can be programmed to be output by the simulator. The NVIDIA Jetson Texas model contains the software for the neural network and reinforcement learning algorithms employed to implement the decision controller. Machine learning frameworks like TensorFlow and GPU frameworks like CUDA are installed on this device. The datasets used in this study were obtained from the Waymo Open Dataset, available at https://github.com/waymo-research/waymo-open-dataset/tree/master/src/waymo_open_dataset.

Experimental evaluation

The experimentation part adopts a platoon situation in which an autonomous vehicle follows the master conventional vehicle. The autonomous vehicle will be the primary objective for assessment reasons. The autonomous vehicle setup consists of an engine control unit with detectors like lidar technology, orientation and velocity sensors, and controllers like accelerator and steering. As a decision controller for the vehicle, DDQN receives inputs such as sensing detectors and output actuation to control the vehicle’s propulsion and steering. Figure 4 shows sample images from the simulated environment.

Figure 4 Sample images from simulated environment.

The objective of this experimentation is to maintain a steady pace and separation from the master vehicle, all while keeping the vehicle in its secure position in the middle of the roadway. Several measures, such as controller activation periods, turnaround times such as the availability and downtime of the system, the position of the autonomous vehicle, and proximity from middle of road are examined to evaluate the impact of the proposed system. The decision controller used in the implementation is constructed as a model with sequences. The model generates a vehicle management sequence with the acceleration of the vehicle and steering readings as its output after receiving an input vector containing the results from nine lidar imaging detectors, speed, brake, and gear. Ten hours of conventional automobile driving information from the simulator are used to train this model. The model generates a reliable foundation operational environment by simulating an automobile that follows the master car at a speed of about eighty miles per hour.

The experimental results produced by the proposed system in terms of collision rate, similarity in behavior, distance, speed, total rewards produced, and time steps taken are presented in Figs. 5, 6, 7, 8, 9, 10. The potential of an agent to minimize the collision rate is assessed as one of the most important aspects. The frequency with which the agent fails to avoid crashes while driving is shown by the collision rate. Better performance is indicated by a lower collision rate value and can be mathematically represented as shown in Eq. (15)

(15) CR=TcollisionsTruns

where Tcollisions denotes the total number of collision events, and Truns denotes the total number of simulated runs.

Figure 5 Collision rate.

Figure 6 Similarity in behaviour.

Figure 7 Distance covered.

Figure 8 Speed regulation.

Figure 9 Total rewards produced.

Figure 10 Timesteps taken.

When it comes to autonomous driving, a collision indicates an operational error on the part of the automotive CPS, which can lead to disastrous results. Hence, the agent is trained extensively in multiple simulated settings that replicate conventional manual driving scenarios adopted from the master vehicle to attain a low collision rate. The agent is put through various scenarios in these simulations to improve its reasoning processes by continuously learning from these encounters, which enables it to foresee possible collisions and take preventative measures to avoid them.

The degree to which the agent’s driving behaviours resemble those of manual driving is another crucial factor in assessing the agent’s performance.

The performance that the agent mimics needs to be evaluated to evaluate the performance of any living being, for, in this case, it is manual driving. The measure used to ensure that the agent’s analogues perform as they are supposed to is called a behavioural similarity. As far as driving involves vehicle control actions like pedal and steering wheel thrusting, which includes braking, accelerating, or changing the steering angle, this can be distilled from the actions of the agent and that a real pilot makes. Mathematically, this is expressed as Eq. (16).

The degree to which the agent’s driving behaviours resemble those of manual driving is another crucial factor in assessing the agent’s performance. The agent’s driving habits are guaranteed to mimic those of a human driver according to the behavioural similarity metric. This can be measured by comparing the agent’s and a human driver’s control actions, such as braking, acceleration, and steering angle. This can be mathematically formulated as shown in Eq. (16),

(16) BS=1−1T∑t=1T|atagent−athumandriver|.

In the above equation, T denotes the total number of timesteps used in the simulation, atagent denotes the action taken by the autonomous agent at timestep t, and athumandriver denotes the action taken by the human driver at timestep t.

An autonomous vehicle needs to be highly secure and effective, but it also needs to act in a way that makes sense for human drivers and occupants. The agent’s behaviour in the simulated environment is guaranteed to be in line with the behaviour of conventional driving agents. The agent is trained to abide by standard driving procedures, including signalling before lane changes, keeping a safe distance from other cars, and observing speed limitations. These actions improve safety and facilitate the integration of autonomous vehicles into traffic networks.

The third important measure that is assessed is the distance that the autonomous vehicle travels in a given amount of time. Distance refers to both the distance a vehicle can go and the efficiency with which it can arrive at its intended location. Mathematically represented as given in Eq. (17),

(17) DV=∑t=1Trt⋅Δt.

In the above equation, rt denotes the speed, and Δt denotes the duration of each timestep. The ability to design and carry out the most efficient routes while considering variables like the volume of traffic, roadway conditions, and barrier existence should also be incorporated into an optimized reinforcement learning agent, which can be considered for extended research in the future.

Speed has a crucial role in determining the performance and safety of autonomous vehicles. Consistent speed is essential for both efficiency and safety. The standard deviation of speed can be used to quantify speed variation and formulated mathematically as given in Eq. (18),

(18) Rc=1T∑t=1T(rt−r¯)2.

In this equation, (r¯) refers to the average speed throughout the episode. A lower Rc means less fluctuation in speed changes, which restrains unnecessary braking and acceleration.

The agent is evaluated based on the speed shift regarding the observed roadway, other vehicles, and the posted speed limit and is subsequently rewarded. The total rewards issued to autonomous vehicles are associated with driving performance, which prevents accidents, provides aid for adequate movement, and complies with traffic regulations.

The agent’s adherence to driving policies is gauged by the total incentives earned during an episode. The total rewards aggregated ( Ragg) are computed based on the reward function defined for each state-action pair ( RF(s,a)).

(19) Ragg=∑t=1TRF(s,a)

(20) RF(s,a)=v1⋅RWsafe+v2⋅RWspeed+v3⋅RWlegitimate+v4⋅RWefficiency.

In the above equation, RWsafe denotes the reward for collision avoidance, RWspeed denotes the reward for maintaining optimal speed, RWlegitimate denotes the reward for valid path changes, RWefficiency denotes the reward for reaching the goal in minimal time, and v1,v2,v3, and v4 are the weights that can be adjusted to balance these rewards.

By continually choosing actions that are in line with its training targets, an optimized automotive CPS will seek to maximize its aggregate rewards. Positive rewards are given for avoiding collisions, observing speed constraints, and choosing safer paths. On the other hand, activities that incur negative rewards, such as causing a crash or breaking traffic rules, will lower the total rewards. The rewards are assigned as per the details specified in “Identifying Indeterminate States” under the scheme for rewarding mechanisms.

The efficiency of autonomous driving is also assessed based on the number of timesteps taken to finish a driving task. A timestep in reinforcement training denotes a single instance in which a learning agent perceives the surroundings, decides what to do, and acts upon that decision. Fewer timesteps will be required for an optimised agent to do tasks, suggesting that it can make speedy decisions. Hence, the agent in the simulated environment is also assessed based on this perspective.

Limitations of current research

Despite being trained in simulated contexts, the reinforcement-based agent might not be able to extrapolate well enough to function at its best in real-life scenarios. Unfavourable behaviour or failure in unexpected situations could result from differences between modelled training scenarios and real-world circumstances, such as unanticipated impediments, intricate traffic patterns, or uncommon, rare cases. It takes a lot of labour and computational capacity to accomplish optimised behaviour through the considerable training that is essential. It is difficult to scale this training to encompass a wide range of driving scenarios, such as different types of roads and traffic patterns.

Strategies like domain randomization, which changes simulations such as lighting, road textures, and the actions of pedestrians randomly during the training process to increase the agent’s adaptability, may be used to overcome the model’s restrictions. Another approach to this problem is using domain invariant feature learning, which instructs the machine to learn standard features in simulation and actual world data. The RL agent’s effectiveness can be enhanced with simulation pretraining followed by a small amount of actual data for fine-tuning. The agent can rapidly adapt to new and previously unseen conditions by employing meta-learning.

Conclusion

The proposed research solves the problems posed by such driving environments by incorporating the DDQN into automobile cyber-physical systems. The study proposes new ways to enhance the decision-making processes of autonomous vehicles by devising the motivational framework and modelling the outcome of external shocks. In addition to modification of the system’s flexibility under dynamic driving conditions, this systematic approach gives a better understanding of how reinforcement learning’s reward structure works. This work grapples with the problem of what adaptable and compact hardware architectures are needed to enable the aforementioned complex algorithms. Integrating field programmable gate arrays as a hardware solution is beneficial due to its ability to optimise trade-offs between post-manufacturing modification and parallel computing and significant performance parameters. These factors that allow FPGA-based implementations to increase computational efficiency, reliability, and flexibility ensure a better autonomous driving framework through a more complex algorithm.

Various metrics, such as collision rate, behavioural similarity, journey distance, speed regulation, total rewards, and timesteps, help evaluate the system’s performance. According to simulation tests conducted in the TORCS Racing Simulator, the proposed method surpasses the engineering constraints of automobile CPS. It simultaneously improves the efficacy, trustworthiness, and real-time decision-making in autonomous driving systems. To enhance the generalisation ability of the agent to novel environments, future work should focus on exploring self-supervised and meta-reinforcement learning hybrid approaches. Moreover, to further narrow the gap between the driving performance in the simulation and the real world, adversarial training and domain randomization should be incorporated.

Additionally, multi-agent reinforcement learning (MARL) can further enable cooperation among autonomous vehicles in complex traffic situations, increasing overall safety and efficiency. In a nutshell, this is research emerging in the realms of hardware-efficient real-time reinforcement learning for autonomous cars, which makes self-driving cars unprecedentedly safe and flexible. The work can make further strides in intelligent autonomous driving systems by overcoming existing limitations and pursuing the suggested future research directions.

Supplemental Information

Supplemental Information 1 Dataset.

Supplemental Information 2 Implementation code.

Additional Information and Declarations

Competing Interests

The authors declare that they have no competing interests.

Author Contributions

Manal Abdullah Alohali conceived and designed the experiments, analyzed the data, authored or reviewed drafts of the article, and approved the final draft.

Hamed Alqahtani conceived and designed the experiments, performed the experiments, analyzed the data, performed the computation work, authored or reviewed drafts of the article, and approved the final draft.

Abdulbasit Darem conceived and designed the experiments, analyzed the data, authored or reviewed drafts of the article, and approved the final draft.

Monir Abdullah performed the experiments, performed the computation work, prepared figures and/or tables, authored or reviewed drafts of the article, and approved the final draft.

Yunyoung Nam performed the experiments, performed the computation work, prepared figures and/or tables, and approved the final draft.

Mohamed Abouhawwash performed the experiments, performed the computation work, prepared figures and/or tables, and approved the final draft.

Data Availability

The following information was supplied regarding data availability:

The data is available at Zenodo: Manal Abdullah Alohali. (2025). Integrating Cyber-Physical Systems with embedding technology for controlling autonomous vehicle driving [Data set]. Zenodo. https://doi.org/10.5281/zenodo.14977858.

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
