# Peer review of "Integrating cyber-physical systems with embedding technology for controlling autonomous vehicle driving"

_PeerJ Computer Science, doi:10.7717/peerj-cs.2823_

## Round 0.1 · original submission · Minor Revisions

Dear author,

Your paper has been reviewed by two reviewers. They provided some comments for improvement of the paper. Please make revisions with mark all changes and provide cover letter with replies point to point to their comments.

·

Basic reporting

1. Basic reporting
Clear and unambiguous, professional English used throughout
The authors have demonstrated a good understanding of the academic English language, as a base for the successful application of the mentioned language in writing scientific and professional papers in the field of technical sciences. Furthermore, the tense choice and the style of writing are directed towards the clarity, preciseness, objectivity and analyticity of this article. Checking the entire paper, (some words are repeated - line 67, line 348, line 376), is strongly recommended.
Literature references, sufficient field background/context provided
The introduction consists of three parts: The first one, in which the detailed background of the paperwork is presented. Due to the applied step-by-step approach, the authors gradually introduce the reader to the gaps in current research, (supported by well-chosen reference literature), and on that base the necessity for their overcoming. The mentioned leads to the bulleted list of the research contributions, (1.1.), as well as the organization of the paper, (1.2). In addition, in such a way of preparing the introduction, the authors’ motivation and the goal of the paper are self-imposed, which means they are clear even at the beginning of the article.
Professional article structure, figures, tables. Raw data shared
The paper consists of sections and sub-sections which are in line with the PeerJ standards, as well as, the topic of the research, applied methodology, and the obtained results. There are ten figures and one table in the frame of the article. They are clear, (except Figure 1, which should be supplied with a higher quality), there is a caption at the bottom/top of each figure and presented table. Some of them, (Figure 1,2,3,4,5,6) are placed in the appropriate position within the text, to additional explain, clarify or draw attention to particular parts of the content. This does not apply to other figures, (7,8,9,10). Therefore, the mentioned should be corrected. Raw data are provided. (Lines 345-346). They are accessible in supplemental files.
Self-contained with relevant data to the hypothesis
This article is well conceived. It means that all parts are written in such a manner that is easy to understand their contents, without, (at the beginning of the review), a deeper analysis of the next sections. Other chapters of the article contain data oriented towards proving the hypothesis set in the introduction, (Lines 108 – 119).
Formal results should include clear definitions of all terms and theorems and detailed proofs.
Terms and theorems are defined and explained in different parts of the paper, (some are supported with relevant references), depending on the purpose. Results are confirmed and presented using figures in subsection 4.2.

Experimental design

Original primary research within Aims and Scope of the journal
This research article is in the field of Computer Sciences. The authors follow the link titled Scope of the journal, such as parts refer to aims & scope, discipline specific standards, instructions for authors, research manuscript template etc.
Research question well defined, relevant & meaningful. It is stated how the research fills an identified knowledge gap
This request is fulfilled. Namely, the relevant reasons for researching the mentioned topic are described in detail one by one in the introduction of the paper. At the end of the section, the authors present the way to overcome the problems. i.e. making “decision controllers for autonomous driving CPS in an indeterminate context, where the learning process is directed by anticipated cognitive restrictions.” (Lines 109 - 111). In addition, using of bullet formats is oriented towards a clear and concise presentation of the research contribution.
Rigorous investigation performed to a high technical & ethical standard
The organization of the investigation is based on technical standards, such as the definition of terms, (presented in particular sections depending on the type and their purpose in the research), a precise description of the procedure, (Lines 108 – 119), detailed characterization of the applied methodology, as well as a rationale for the chosen approach - section 3, test methods, (sub-section 4.2).
Ethical standards are also implemented. Declaration of the authors that they have no competing interests, declaring the external funding sources, agreement of the authors regarding the accountability of all parts of the investigation, approving the final draft of the manuscript submitted for review, etc. are used in the process of investigating the problem.
Methods described with sufficient detail & information to replicate
The methodology applied for the investigation is presented in detail in section 3 of the paper. Therefore, at the beginning of this part, the authors explained the scientific base for using the proposed methodology. Also, there is an additional explanation of the proposed system in Figure 1. Moreover, it should be emphasized that there are subsections related to the methodology, which are in connection with the subject and main goal of the research.
Clear, specific and complete information about the available data, applied methodology and computational environment is a base for ensuring the reproducibility of the obtained results. (section 3 and section 4). Moreover, the authors provide an appropriate description of the relevant limitations of their research. (non-replicability, subsection 4.3). The mentioned can lead to new insights and increased knowledge of the research topic.

Validity of the findings

Impact and novelty are not assessed. Meaningful replication is encouraged where rationale & benefit to literature are clearly stated
The authors of this article introduce a new approach in the investigation of automotive cyber-physical systems’ possibilities based on the AI algorithms application. The mentioned is explained in the sections and subsections of the article, especially in 1Introduction, (Lines 108 – 118), as well as, 1.1. Research contributions, (Lines 120 – 128).
The meaningful replication, (“Replication is as much a part of the scientific method as formulating a hypothesis or making observations.”) available at Why is Replication in Research Important? | AJE is possible, based on section 3 and section 4.
All underlying data have been provided; they are robust, statistically sound & controlled
The underlying data are provided. (section 4 – Lines 345 and 346). They are comprehensive with a wide area of applications, controlled and statistically significant.
Conclusions are well stated, linked to the original research question & limited to supporting results
Conclusions are prepared as summaries of the main points of the article, such as the subject of investigation, proposed methods and reasons for their use (in connection with the goal of the paper), and the main findings. It is recommended that conclusions be prepared in the form of a listing.
To complete this part of the article, the remaining problems, questions, and areas for future research need to be pointed out. Namely, limitations of current research as well as recommendations for further works are presented in the part that refers to Results and discussion. Besides the fact that the authors paid attention to the directions for further study, there is a need for a deeper analysis of the factors that are the basis for additional research. That will be a significant contribution towards a closer and more focused examination of the paper’s topic.

Additional comments

4. General comments
The article titled “Integrating Cyber-Physical Systems with embedding technology for controlling autonomous vehicle driving” is an original paper in the field of Computer Science.
Given artificial intelligence’s profound impact on daily life, this technology is involved in many topical research works.
Therefore, it can be concluded that the above-mentioned paper related the autonomous vehicles, (as a product of processes of taking the human element out of transportation), is characterized by a high level of actuality.
In addition, it is essential to point out that the investigation which takes into account this new type of vehicle makes a significant contribution in the fields of Traffic and Transport, Mechanical Engineering, Civil Engineering, Transport and Environment, as well.

·

Basic reporting

Reviewer Comments for Manuscript
" Integrating Cyber-Physical Systems with embedding technology for controlling autonomous vehicle driving"

Overall, the manuscript addresses a significant and challenging topic. The strategy adopted to tackle the issue is appropriate, and the results could be of interest to readers. The author should address the points mentioned below to improve the overall quality and impact of the paper, and I recommend the manuscript for publication after addressing all the comments below.
• The abstract should highlight the uncertainty faced during modeling to effectively address the application of CPSs.
• Add keywords that are not part of the title.
• Write in a subsection the statement of the research problem, its importance and necessity, a review and critique of the current state, identification of research gaps, and an indication of the study's innovation and contribution must be mentioned.
• Citations are comprehensive, but it would be beneficial to include more recent references to show the current relevance of the research and some literature review in the sense of CPSs, indeterminacy, and autonomous vehicles.
• Add data statement and the pictures used in the testing of the modeling.
• Figures 3-10 are not cited in the text properly cite them.
• Broaden the future directions to utilize these results in different modeling scenarios to emphasize the study's implication.
• Add sensitivity analysis of the proposed work or do a necessary comparison.
• Refine language, grammar, and formatting for better clarity and consistency.

Experimental design

The author should address the points mentioned above to improve the overall quality and impact of the paper, and I recommend the manuscript for publication after addressing all the below comments.

Validity of the findings

The author should address the points mentioned above to improve the overall quality and impact of the paper, and I recommend the manuscript for publication after addressing all the below comments.

Additional comments

The author should address the points mentioned above to improve the overall quality and impact of the paper, and I recommend the manuscript for publication after addressing all the below comments.

---

## Round 0.2 · accepted · Accept

Dear authors,

Your paper has been reviewed in the second round of review and we are pleased to inform you about a positive decision. Reviewers have recommended acceptance of your paper.

·

Basic reporting

Thank you for providing the revised version of the manuscript. I have carefully reviewed the authors' revisions and responses to the comments. I am satisfied with the changes made, as they have adequately addressed the concerns raised in the previous review.

I recommend accepting the manuscript for publication in its current form.

Experimental design

Thank you for providing the revised version of the manuscript. I have carefully reviewed the authors' revisions and responses to the comments. I am satisfied with the changes made, as they have adequately addressed the concerns raised in the previous review.

I recommend accepting the manuscript for publication in its current form.

Validity of the findings

Thank you for providing the revised version of the manuscript. I have carefully reviewed the authors' revisions and responses to the comments. I am satisfied with the changes made, as they have adequately addressed the concerns raised in the previous review.

I recommend accepting the manuscript for publication in its current form.

Additional comments

Thank you for providing the revised version of the manuscript. I have carefully reviewed the authors' revisions and responses to the comments. I am satisfied with the changes made, as they have adequately addressed the concerns raised in the previous review.

I recommend accepting the manuscript for publication in its current form.